# Synthesis and the Crystal Structure of a New Chiral 1D Metal–Organic Coordination Polymer Based on L-Prolineamide-Substituted Diarylacetylenedicarboxylic Acid Derivative

**DOI:** 10.3390/molecules27238376

**Published:** 2022-11-30

**Authors:** Vladimir V. Veselovsky, Vera I. Isaeva, Vera D. Nissenbaum, Leonid M. Kustov, Vladimir V. Chernyshev

**Affiliations:** 1N.D. Zelinsky Institute of Organic Chemistry, Russian Academy of Sciences, Leninsky prospect 47, 119991 Moscow, Russia; 2Laboratory of Nanochemistry and Ecology, National University of Science and Technology MISiS, Leninsky prospect 4, 119991 Moscow, Russia; 3Department of Chemistry, M.V. Lomonosov Moscow State University, 1–3 Leninskie Gory, 119991 Moscow, Russia; 4A.N. Frumkin Institute of Physical Chemistry and Electrochemistry, Russian Academy of Sciences, Building 4, 31 Leninsky prospect, 119071 Moscow, Russia

**Keywords:** homochiral metal–organic coordination polymer, powder X-ray diffraction, chiral diarylacetylenedicarboxylic acid derivative

## Abstract

The new homochiral 1D metal–organic coordination polymer [**Cu_2_(EDPB)•H_2_O]_n_** was synthesized starting from the original 3,3′-ethyne-1,2-diylbis[6-(L-prolylamino)benzoic acid] (H_4_EDPB). The unique crystal structure of the new compound was established by powder X-ray diffraction. The [**Cu_2_(EDPB)•H_2_O]_n_** system shows catalytic activity and enantioselectivity in a Henry reaction of *p*-nitrobenzaldehyde with nitromethane.

## 1. Introduction

For the last thirty years, investigations related to metal–organic coordination polymers (MOCPs) of different dimensionality (one- (1D), two- (2D) and three- (3D) dimensional networks), composed of metal ions bridged by polytopic organic ligands (linkers), have been extremely popular [1]. The reason for this relates to the unusual physico-chemical properties of these highly ordered crystalline materials, which are common in some aspects, and differ in relation to their topology or architecture. One-dimensional MOCPs are linear polymers, which crystallize as chains [2]. Two-dimensional MOCPs exist mostly as layers. Three-dimensional MOCPs, or metal–organic frameworks (MOFs), are extensively studied, due to their extremely high porosity and specific surface area, which provide their potential as multifunctional materials with diverse applications [3].

Despite their structural simplicity, compared to MOFs, 1D MOCPs are also attractive objects for investigation, because they can serve as models for investigation of the correlations between synthesis variables and formed MOCP dimensionality [4,5], and thereby for developing a strategy for the preparation of coordination compounds of desirable dimensionality [6]. Another correlation, which can be studied using 1D MOCPs as models, is the relationship between MOCP dimensionality and its functional properties [7,8,9]: for instance, 1D and 2D MOCPs are rather promising as components of heterogeneous catalysts, because their architectures provide accessibility to the active sites of the substrates [10,11].

It is well known that 1D, 2D and 3D MOCPs can be obtained from the same organic (linkers) and inorganic (metal ions) building blocks, by spatially changing the synthesis variables. Understanding the possibility of self-assembly of the specific coordination network in a predictable way is important for the functional material design. For the construction of MOCPs of different dimensionality, including 1D networks, the most popular organic building blocks are arenedicarboxylic [12,13] and diarylacetylenedicarboxylic [14,15] acid derivatives in combination with transition metal ions (Zn^2+^, Cd^2+^, Co^2+^, Fe^2+^, etc.): the chiral derivatives of these acids have particular potential, because they pave the way to the enanthioselective functional materials including 1D MOCPs with specific properties [13,16].

We have previously reported the synthesis of diarylacetylenedicarboxylic acid [3,3′-ethyne-1,2-diylbis[6-(L-prolylamino)benzoic acid, (EDPBA) (**1**)], which can form the homochiral Hydrogen-Bonded Organic Framework ZIOC-1 by self-assembly [17]. The ZIOC-1 material was studied as an enantioselective heterogeneous catalyst with Lewis basicity in the aldol reaction and Michael condensation. In this context, the next step of our investigations was to study the possibility of dicarboxylic acid **1** utilization as an organic linker for the self-assembly of the chiral MOCPs with Lewis active sites for enantioselective catalytic reactions involving the C–C bond formation.

Herein, we present the results of our study to determine the possibility of synthesizing MOCPs based on building block **1**. It was found that the reaction of EDPBA (**1**) with Cu(OAc)_2_•H_2_O smoothly proceeded in diluted aqueous ammonia at 80 °C, affording the novel 1D MOCP with formula [**Cu_2_(EDPB)•H_2_O]_n_**, where H_4_**EDPB**=EDPBA, along with NH_3_ removal from the reaction medium. The crystal structure of this novel 1D MOCP was established by the powder XRD analysis, its thermal stability was studied by thermogravimetric analysis (TGA), and its morphology was determined by FE-SEM measurements. It is well known that a number of Cu(II) complexes are effective as homogeneous [18,19,20] or heterogeneous catalysts [21] in Henry reactions (nitroaldol condensation). We intended to check the assumption that the [**Cu_2_(EDPB)•H_2_O]_n_** material would operate as a catalyst in this process, due to Lewis base sites in amide moiety and Lewis acid sites as Cu^2+^ ions. These bifunctional properties were observed by Pombeiro et al. [22] for Zn, Cd and Co MOCPs of different dimensionality; therefore, an evaluation of the catalytic activity and enantioselectivity of the novel chiral 1D MOCP was carried out on the enantioselective Henry reaction between *p*-nitrobenzaldehyde and nitromethane.

## 2. Results and Discussion

### 2.1. Crystal Structure

The crystal data, data collection and refinement parameters for new polymer [**Cu_2_(EDPB)**•**H_2_O]_n_** are given in Table 1, and diffraction profiles after the final bond-restrained Rietveld refinement are shown in Figure 1 (see also Appendix A).

The asymmetric unit of [**Cu_2_(EDPB)•H_2_O]_n_** contains half of the chiral ligand EDPB, one Cu(II) ion and one water molecule situated on a 2-fold symmetry axis. Each copper center is coordinated by one O and two N atoms from one EDPB and one O atom from another EDPB, and these four atoms form a basal plane with Cu-(O,N) bond lengths covering the range 1.935(6)–2.053(6) Å. Two O atoms are also involved in the coordination of the close symmetry related Cu(II) ion, thus participating in the formation of the binuclear core (see Figure 2 prepared with *Mercury* [23]) with Cu…Cu separation of 3.050(3) Å. The water molecule (O4) situated on a 2-fold axis weakly coordinates both the symmetry-related copper centers [Cu1…O4 2.481(7) Å], thus complementing the coordination environment of each center with the heavily distorted square-base pyramid (see Appendix A).

In the chiral ligand EDPB, two amide groups (N1) and two hydroxy groups (O1) are deprotonated, so the EDPB has negative charge −4. All bond lengths and angles in the EDPB lie within normal ranges, comparable with those found in the Cambridge Structural Database (ConQuest version 2.0.2 with updates) [24] for close compounds. Two benzene rings make a dihedral angle of 24.7(9)°. In the crystal, each L-proline fragment of L coordinates is in a tridentate mode, thus providing a polymeric chain formation, extended in *101* (Figure 3), and consisting of alternating [2Cu.H_2_O]^4+^ binuclear fragments and L^4−^ bridging ligands. The N2–H2 group is involved in the hydrogen bonding with the O2 atom within this linear polymeric chain (Table 2), while the water molecule participates in hydrogen bonding between the polymeric chains (Table 2), thus consolidating the crystal packing along with the weak C—H…O interactions (Table 2, see also Appendix A).

### 2.2. Morphology of the **[Cu_2_(EDPB)•H_2_O]_n_** Material

The morphology of the new polymer was studied by the SEM method. SEM micrographs (Figure 4) show that the **Cu_2_(EDPB)•H_2_O]_n_** material crystallized in the form of thin plates (“petals”, Figure 4 left) with the sizes of ~ 300–400 nm, which were combined in distorted spherical agglomerates (“rose flowers”, Figure 4 right) of 3–5 μm in size. This morphology is rather unusual for 1D coordination polymers. For instance, 1D MOCP with Zn^2+^ ions and pyrazine-2,5-dicarboxylate ligands, as described in [7], crystallizes as large rectangular plastins or prisms with sizes 4–7 μm. On the other hand, the hydrogen-bonded framework **ZIOC-1**, based on the same linker EDPBA, crystallizes as separate thin plates in almost the same shape, but with much bigger sizes of 1–5 μm [17].

### 2.3. Investigation of **[Cu_2_(EDPB)•H_2_O]_n_** Thermal Stability

The thermal stability of the novel coordination polymer was studied by the TGA method. Figure 5 presents a mass loss and, thereby, the [**Cu_2_(EDPB)•H_2_O]_n_** sample thermal stability during programmed heating. In the low temperature field (up to 130 °C), the mass change, which is around 1.9 %, is associated with water loss. In the temperature range 130–240 °C, remarkable mass change is not observed (0.8%). The next stages of the mass changes (38.5%) and (35%), in the temperature ranges 240–325 °C and 240–325 °C, respectively, can be attributed to decomposition beginning. The rapid mass decrease indicates the [**Cu_2_(EDPB)•H_2_O]_n_** structure collapses at temperatures as high as 450 °C.

### 2.4. FTIR Studies of the **[Cu_2_(EDPB)•H_2_O]_n_** Polymer

In the IR spectrum of the [**Cu_2_(EDPB)•H_2_O]_n_** polymer, there are characteristic adsorption bands at 1606 cm^−1^ and 1401 cm^−1^, belonging to the vibrations ν_as_(COO^−^) and ν_s_(COO^−^) of the carboxylate groups coordinated with Cu(II) [25], respectively (Appendix A). In the IR spectrum of the EDPBA linker, there is a characteristic adsorption band at 1686 cm^−1^, belonging to the carboxylic group of the aromatic acid, and the bands belonging to the carboxylate anion are absent (Appendix A).

### 2.5. Evaluation of the Catalytic Activity of **[Cu_2_(EDPB)•H_2_O]_n_** Polymer

Nitroaldol condensation is one of the main important processes in organic synthesis, as a rather simple route for C–C bond formation. The homochiral nature of a new coordination polymer motivated us to explore it as the catalyst in an enantioselective Henry reaction between chiral *p*-nitrobenzaldehyde and nitromethane in a propanol-2 medium.

The catalysis mechanism suggested for the Henry reaction involved the coordination of benzaldehyde and Cu(II) center, which is an initial step for the C–C bond coupling reaction between carbonyl compounds and nitroalkanes [26].

In this test reaction, the [**Cu_2_(EDPB)•H_2_O]_n_** material did not show high activity. The scalemic (non-racemic) product ((4-nitrophenyl)-2-nitroethanol) was isolated with the yield ~15%, after keeping the reaction mixture for 48 h at 50 °C: this low activity was probably attributable to the low acidity of Lewis Cu(II) active sites; a high coordination number (5) participated in the coordination, with two oxygen atoms of carboxylic groups, two nitrogen atoms of amide, and amino groups and a water molecule; therefore, there were almost no vacant sites for the substrate coordination.

It is noteworthy that the 3D MOCP (MOF) material with Cu^2+^-ions studied by us, i.e., the HKUST-1 material based on benzene-1,3,5-tricarboxylate linkers [21], showed almost no activity in the Henry reaction. The lower activity of the porous HKUST-1 material in this process, as compared to the [**Cu_2_(EDPB)•H_2_O]_n_** polymer, could also be explained by the rather high coordination number of the Cu(II) center, and the missing Lewis sites that naturally disliked the 1D counterpart additional-amide Lewis basic sites.

The enantiomeric excess of the reaction product was ~26% ee. The comparison of values of the retention times of the components of the mixture, measured by HPLC in related conditions [27], allowed us to assign (1*S*)- absolute configuration for a main enantiomer (see Appendix A).

Similarly low enantioselectivity of the 2D MOCPs with Zn^2+^, Cd^2+^ and Co^2+^ ions in the Henry reaction was observed in [11]; it is probable that effective enantioselective recognition may only be provided by 3D MOCP material (MOF) with pores containing appropriate functional chiral groups.

This study may help to provide a certain insight into the different impacts of the porous structures of MOCPs on their catalytic activity and enantioselectivity.

## 3. Materials and Methods

All solvents and reagents were purchased from Acros Organics, and were used without further purification. The homochiral acid EDPBA (**1**) was obtained as previously reported [17]. Elemental analysis was performed on a Perkin Elmer 2400 C,H,N analyzer. Thermogravimetric analysis was performed using the Derivatograph-C instrument (MOM Company). FE-SEM measurements with samples coating were carried out using a Hitachi SU8000 field emission scanning electron microscope (FE-SEM). A target-oriented approach was utilized for the optimization of the analytic measurements [28]. Before measurement, the samples were mounted on a 25 mm aluminum specimen stub, and fixed by conductive silver paint. Metal coating, with a thin film (10 nm) of gold/palladium alloy (60/40), was performed, using a magnetron sputtering method, by a high-resolution sputter coater Cressington 208HR. Images were acquired in secondary electron mode at 2−10 kV accelerating voltage, and at a working distance of 8–10 mm. The morphology of the samples was studied, taking into account the possible influence of metal coating on the surface [29]. FTIR spectra were recorded with a Bruker ALPHA-T instrument.

### 3.1. Synthesis of Coordination Polymer [Cu_2_(EDPB)•H_2_O]_n_

The formation of the [**Cu_2_(EDPB)•H_2_O]_n_** is realized according to the Figure 1. Cu(OAc)_2_•H_2_O in 70 mL of water 25% aq. NH_3_ (4.0 mL) was added to a stirred suspension of 330 mg (0.67 mmol) acid **1** and 270 mg (1.35 mmol, ~2.0 equiv.), in order to solubilize it. The resultant solution was stirred in air at 80 °C in an open vial for 2 h, until its pH changed from ~10 to almost neutral, due to NH_3_ removal, by evaporation, from the solution. The resultant powder was separated by centrifugation, and washed with water (4 × 5 mL). Drying of the crude product for 20 h in vacuo under P_2_O_5_ (2 Torr, 20 °C) gave coordination polymer [**Cu_2_(EDPB)•H_2_O]_n_** with a yield of 380 mg (78%). Elemental analysis calculated (%) for C_26_H_24_Cu_2_N_4_O_7_: C 49.44; H 3.83; N 8.87. Found: C 48.82; H 3.69; N 8.74.

### 3.2. Thermogravimetric Analysis

The dried sample (9.4 mg) of [**Cu_2_(EDPB)•H_2_O]_n_** was placed in a platinum crucible, and heated in air from 20 to 500 °C at a heating rate of 5 °C min^−1^.

### 3.3. X-ray Powder Diffraction

The crystal structure of the coordination polymer [**Cu_2_(EDPB)•H_2_O]_n_** was determined by the use of SDPD (structure determination from powder diffraction) methods [30,31,32,33,34]. The X-ray powder pattern was measured at room temperature on a Panalytical EMPYREAN diffractometer with a linear Xcelerator detector, using non-monochromated Cu K_α_ radiation. All observed peaks were indexed in a monoclinic C-centered unit cell with a volume of 1239 Ǻ^3^. The Pawley fitting [35] performed with the program MRIA [36] confirmed the space group *C2*. The crystal structure was solved in this space group with the use of the simulated annealing technique [37], assuming location of the chiral ligand on a 2-fold symmetry axis. The solution obtained in simulated annealing runs was refined with the program MRIA, following the known procedures described by us previously [17,38,39]. A careful inspection of the Fourier difference map allowed us to find, on a 2-fold symmetry axis, the position of the water molecule coordinating the two neighboring Cu atoms. In the refinement, only three isotropic displacement parameters were refined: two parameters for Cu and water’s oxygen, respectively; one common *U_iso_* parameter was used for the rest of the non-H atoms. All the H atoms were placed in calculated positions, and not refined. The strong anisotropy of diffraction-line broadening was approximated by a quartic form in *hkl* [40].

### 3.4. Evaluation of the Catalytic Performance of **[Cu_2_(EDPB)•H_2_O]_n_** Material in a Henry Reaction

The Henry of *p*-nitrobenzaldehyde with nitromethane proceeds according to the Figure 2. A mixture of [**Cu_2_(EDPB)•H_2_O]_n_** (40 mg, 0.063 mmol, 0.1 equiv.), *p*-nitrobenzaldehyde (95 mg, 0.63 mmol) and nitromethane (390 mg, 6.39 mmol) in i-PrOH (1.5 mL) was stirred at 50 °C in an argon atmosphere. The reaction mixture was stirred for 48 h, and was then diluted with EtOAc (10 mL). The catalyst was separated by centrifugation, and was washed with EtOAc (3 × 5 mL). The combined supernatant was concentrated in *vacuo.* The residue was purified by chromatography on SiO_2_. Elution with a mixture of CHCl_3_ gave, in order of elution, unreacted *p*-nitrobenzaldehyde (60 mg) and the non-racemic product of nitroaldol reaction (1-(4-nitrophenyl)-2-nitroethane-1-ol) with a yield of 20 mg (15%) as viscous oil (^1^H NMR (200 MHz, CDCl_3_) *δ* (ppm): 3.47 (br.s, 1H, OH); 4.58–4.62 (m, 2H, CH_2_O); 5.61 (dd, 1H, CHO, *J* = 7.4, 5.0 Hz); 7.62 (d, 2H, 2 HAr, *J* = 8.7 Hz); 8.23 (d, 2H, 2HAr, *J* = 8.7 Hz, cf. [25] (SI). HPLC analysis (Appendix A) of chiral phase Kromasil 3 CelluCoat (column 4.6 × 150 mm), hexane/i-PrOH 90:10: flow rate 1.0 mL/min. The retention times of the enantiomers were 22.80 min for minor and 27.03 min for major (~26% ee).

## 4. Conclusions

A new chiral metal–organic coordination polymer [**Cu_2_(EDPB)•H_2_O]_n_** was constructed in a “green” way by the self-assembly of L-prolineamide-substituted diarylacetylenedicarboxylic acid derivative (EDPB) and Cu^2+^ ions in a water medium, under rather mild synthesis conditions (80 °C, 2 h, air): its crystal structure was established with the use of X-ray powder diffraction. The [**Cu_2_(EDPB)•H_2_O]_n_** material was thermally stable until 240 °C, as well as chemically stable in common solvents, such as chloroform and acetone. The [**Cu_2_(EDPB)•H_2_O]_n_** material showed a certain catalytic activity in a Henry reaction between *p*-nitrobenzalehyde and nitromethane, affording a non-racemic product (4-nitrophenyl)-2-nitroethanol), which was higher than the catalytic activity of the 3D metal–organic framework HKUST-1. It is probable that the catalytic activity of the new 1D MOCP could be ascribed to the presence of the Cu^2+^ ions as Lewis active sites, and to amide functionalities in EDPB linkers as Lewis base sites.

Further adjustment of 1D MOCP structures, for improved catalytic performance in condensation reactions, will be the focus of our future investigations.

## Data Availability

The data presented in this study are available on request the corresponding authors.

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
