# Peer review of "Synthesis and the Crystal Structure of a New Chiral 1D Metal–Organic Coordination Polymer Based on L-Prolineamide-Substituted Diarylacetylenedicarboxylic Acid Derivative"

_molecules, 2022, doi:10.3390/molecules27238376_

Round 1

Reviewer 1 Report

Notes to article “Synthesis and the Crystal Structure of a New Chiral 1D Metal-organic Coordination Polymer based on L-Prolineamide-Substituted Diarylacetylenedicarboxylic Acid Derivative” by authors Vladimir V. Veselovsky, Vera I. Isaeva, Vera D. Nissenbaum, Leonid M. Kustov and Vladimir V. Chernyshev

The article “Synthesis and the Crystal Structure of a New Chiral 1D Metalorganic Coordination Polymer based on L-Prolineamide-Substituted Diarylacetylenedicarboxylic Acid Derivative” by authors Vladimir V. Veselovsky, Vera I. Isaeva, Vera D. Nissenbaum, Leonid M. Kustov and Vladimir V. Chernyshev is devoted to the synthesis and study of an important and interesting class of compounds, namely metal-organic coordination polymers (MOCPs). The undoubted advantage of the work is the use of green chemistry methods to obtain the studied product. The structure determination was carrying out at a high level. It can be seen that the authors of the work are highly qualified specialists in the field of SDPD (structure determination from powder diffraction) methods. Thermogravimetric analysis and evaluation of catalytic activity in Henry reaction for [Cu2(EDPB)•H2O]n were carry out.

The article may be published in the "Molecules" after taking into account some comments.

1.      Authors note that in the crystal hydrogen bonding chains form between the polymeric chains. Drawing is required (it may be placed in Supporting Information file).

2.      Morphology of the [Cu2(EDPB)•H2O]n material is described very briefly. It is necessary to compare the obtained data with data obtained for other similar compounds. Provide relevant links.

3.      Cif-file is necessary to add as Supplementary file.

Author Response

  1. Authors note that in the crystal hydrogen bonding chains form between the polymeric chains. Drawing is required (it may be placed in Supporting Information file).

New Figures S3 & S4 are added to Supplementary and cited in the revised MS.

  1. Morphology of the [Cu2(EDPB)•H2O]n material is described very briefly. It is necessary to compare the obtained data with data obtained for other similar compounds. Provide relevant links.

A short discussion of the [Cu2(EDPB)•H2O]n morphology and its comparison with previously synthesized 1D MOCP with pyrazine-2,5-dicarboxylate linkers and hydrogen-bonded framework ZIOC-1 with the same linker EDPBA was added to the revised MS (P. 5).

  1. Cif-file is necessary to add as Supplementary file.

Done.

Reviewer 2 Report

The paper presents data on the synthesis of a chiral 1D coordination polymer containing copper(II) ions. The authors have characterized the obtained compound by means of X-ray structure analysis, thermogravimetry, SEM pictures demonstrating the morphology of the substance. In addition, the authors showed the catalytic activity of the resulting complex in the Henry reaction. After reading the manuscript some questions and comments arose that would improve the quality of the article.
1) Figure 2. Correct the chain fragment of the coordination polymer so that the two sides of the ligand are symmetrically represented by the bidentate copper fragments.
2) Add "1D chain" to the caption of Figure 3.
3) Correct "GoF" to GooF in Table 1.
4) Add the calculated PXRD curve without subtraction.
5) Please use the calculation with SHAPE software to correctly attribute the shape of the polyhedron of copper ions.
6) In the experimental part. Why is 1.5 equivalent of copper acetate and not 2 equivalent based on the number of moles? please check. In addition it is better to present the equation of reaction with coefficients.
7) What is the supposed effect of the copper complex on the selected organic reaction? (include references)
8) The aldehyde series needs to be extended to demonstrate the scope.

Author Response

1) Figure 2. Correct the chain fragment of the coordination polymer so that the two sides of the ligand are symmetrically represented by the bidentate copper fragments.

Figure 2 is properly corrected.

2) Add "1D chain" to the caption of Figure 3.

Done.

3) Correct "GoF" to GooF in Table 1.

Done.

4) Add the calculated PXRD curve without subtraction.

New Figure S1 showing separately experimental and calculated curves is included in Supplementary and cited in the revised MS.

5) Please use the calculation with SHAPE software to correctly attribute the shape of the polyhedron of copper ions.

New Figure S2 showing the coordination polyhedron of copper ions is included in Supplementary and cited in the revised MS.

6) In the experimental part. Why is 1.5 equivalent of copper acetate and not 2 equivalent based on the number of moles? please check. In addition it is better to present the equation of reaction with coefficients.

Sorry for this mistake. The proper “2 equivalent” was inserted in the synthesis procedure description in the Revised Manuscript (P. 7). In our opinion, in case of MOCP formation, the reaction equation is not properly fit, because reaction product is a macromolecular compound.

7) What is the supposed effect of the copper complex on the selected organic reaction? (include references) 

The relevant consideration of the Henry reaction mechanism along with an appropriate citation of relevant new Ref. 26 were added to the Revised Manuscript (P. 6).

8) The aldehyde series needs to be extended to demonstrate the scope.

 To study the catalytic efficiency of the [Cu2(EDPB)•H2O]n polymer in nitroaldol condensation, we chose 4-nitrobenzaldehyde as the nitromethane acceptor, which exhibits the highest reactivity in this reaction as a rule. In our case a not sufficient yield of the product and its low enantiomeric homogeneity were obtained. Therefore, we intend to modify this MOCP structure and composition in order to improve its catalytic activity/enantioselectivity before exploring the scope and limitations of Henry reaction with other less active aldehydes. This is a topic of our current research efforts.

Reviewer 3 Report

This work by Veselovsky and colleagues describes the synthesis and characterization of a copper-organic coordination polymer having as ligand a diarylacetylenedicarboxylic acid functionalized with L-prolineamide. Noteworthy, the catalytic outcome of an asymmetric Henry reaction was evaluated using the coordination polymer as catalyst.

There is only one minor point that the authos should address in ansy subsequent revision: FTIR measurements of the ligand and the coordination polymer should be performed.

Author Response

1). There is only one minor point that the authos should address in ansy subsequent revision: FTIR measurements of the ligand and the coordination polymer should be performed.

The FTIR measurements were carried out for EDPBA linker and [Cu2(EDPB)•H2O]n polymer. A short discussion of the obtained results was added to the revised MS (P. 6) along with the Figures S5 and S6 added to Supplementary.

Round 2

Reviewer 2 Report

In my opinion, the article looks much better after the corrections made by the authors in the revised manuscript.
I can recommend this paper for publication in Molecules.